# Benzothiazoles from Condensation of *o*-Aminothiophenoles with Carboxylic Acids and Their Derivatives: A Review

**DOI:** 10.3390/molecules26216518

**Published:** 2021-10-28

**Authors:** Efrén V. García-Báez, Itzia I. Padilla-Martínez, Feliciano Tamay-Cach, Alejandro Cruz

**Affiliations:** 1Instituto Politécnico Nacional-UPIBI, Laboratorio de Química Supramolecular y Nanociencias, Av. Acueducto s/n, Barrio la Laguna Ticomán, México 07340, DF, Mexico; efren1003@yahoo.com.mx (E.V.G.-B.); ipadillamar@ipn.mx (I.I.P.-M.); 2Laboratorio de Investigación de Bioquímica, Sección de Estudios de Posgrado e Investigación, Escuela Superior de Medicina, Instituto Politécnico Nacional, Plan de San Luis y Salvador Díaz Mirón s/n, Casco de Santo Tomás, México 11340, DF, Mexico; ftamay@ipn.mx

**Keywords:** *o*-aminothiophenol, carboxylic acid and derivatives, 2-substitutedbenzothiazoles, acid catalysis, microwave assisted, condensation reaction

## Abstract

Nowadays, organic chemists are interested in the field of heterocyclic chemistry due to its use in the synthesis of a great variety of biologically active compounds. Heterocyclic compounds are widely found in nature and are essential for life. Among these, some natural nitrogen containing heterocyclic compounds have been used as chemotherapeutic agents. Their attachment to sugar molecules either as thioglycosides or as nucleosides analogues plays an important role in vital biological processes as well as in synthetic organic chemistry. Molecules containing benzothiazole (BT) nuclei are of this interesting class of compounds because some of them have been found to have a wide variety of biological activities. In this sense, we selected this topic to review and to then summarize the procedures related to the condensation reactions of *o*-aminothiophenoles (ATPs) as well as their disulfides with carboxylic acids, esters, orthoesters, acyl chlorides, amides, and nitriles. The condensation reactions with carbon dioxide (CO_2_) are included. Conventional methods with the use of acid and metal catalysts as well as recent green techniques, such as microwave irradiation, the use of ionic liquids, and ultrasound (US) chemistry, which have proven to have many advantages, were found in the review.

## 1. Introduction

Benzothiazole (BT) belongs to the family of benzazole compounds. The base structure of BT consists of a benzene ring fused with the 4 and 5 positions of a thiazole ring. These two rings together constitute the bases of the planar structure of the BT nucleus. BT is a colorless and slightly viscous liquid compound with a molecular formula C_7_H_5_NS. This is considered as a weak base with a melting point of 2.0 °C, a boiling point of 227–228 °C, a density of 1.24 g/mol, and a molecular mass of 135.19 g/mol. This privileged bicyclic ring system has been found in various natural compounds, which have a wide range of pharmaceutical applications [1,2,3]. Scientists have discovered the BT nucleus in natural products such as bisabolane-type sesquiterpenoid from the roots of *Ligularia dentate* (compositae) and diterpenes erythrazoles A and B of mangrove sediments (Figure 1) [4,5].

BT heterocyclic compounds are rarely found in alkaloids, terrestrial, and marine natural products. However, the compounds found range from docile to a broad range of more complex structures created by the substitution of several groups at the carbon site between the nitrogen and sulfur atoms. These derivatives range in complexity from the well-known simple-structured firefly *d*-luciferin, which was isolated in the late of 1940s [6,7,8], to more complex molecules, such as rifamycins P and Q [9] and thiazinotrienomycin F and G (Figure 2) [10]. Some BT derivatives are also known to be aroma constituents of tea leaves and cranberries or flavor compounds such as the alkaloids nordercitine **Ia**, dercitamine **Ib**, dercitamide **Ic**, and cyclodercitin **Id** produced by the fungi *Aspergillus clavatus* and *Polyporus frondosus* (Figure 2) [1,11]. In addition, since 1950, medicinal chemists have been interested in BT derivative compounds, since according to its pharmacological profile, riluzole (6-trifluoromethoxy-2-benzothiazolamine) was found to be a clinically anticonvulsant drug (Figure 2) [12,13].

BTs have been found to act as core nucleus in various synthesized drugs. For instance, the substituted ones are present in many pharmaceuticals that exhibit remarkable biological and therapeutic activities. For example, since 1991, Zopolrestat (Figure 3**a**) has been used to treat diabetes [14], while (GW610, NSC 721648) (Figure 3**b** and its derivatives show excellent antitumor activity [15,16], and Schiff bases (Figure 3**c**) are used as amyloid inhibitors for the treatment of Alzheimer’s disease [17]. Moreover, the BT moiety is also found in many functional molecules, such as ratiometric fluorescent pH indicators and ligands for catalytic reactions [18,19].

Since 1991, BT has been a moiety known to play an important role in chemistry, biochemistry, and medicinal chemistry, with a wide array of interesting biological activities and therapeutic functions, such as antimicrobial [20,21,22,23], anticancer [24,25,26,27,28,29], anthelmintic [30], antidiabetic [31,32], antituberculotic [33,34], antitumor [35,36,37,38,39,40,41], antitrypanosomal [42], antiviral [43,44,45], antibacterial [46,47,48,49], antioxidant [50], antiglutamate and antiparkinsonism [51,52], analgesic [53], anti-inflammatory [54,55], antifungal [56,57], antileishmanial [58], anticonvulsant [59], neuroprotective [60], muscle relaxant [61], vasodilator [62], and orexin receptor [63]. In addition, it acts as an inhibitor of several enzymes [14,64]; helps in atherosclerosis [65], insomnia [66], and epilepsy [67]; functions as an LTD4 receptor antagonist [68,69]; has antiparasitic and photosensitizer function [70]; and acts as a calcium channel antagonist [71,72], as an imaging agent for Aβ plaques in cerebral amyloid angiopathy [73], and as a falcipain inhibitor [74]. In addition, BT has other applications, e.g., as a radioactive β-amyloid imagining agent [75,76,77,78,79] and a schistosomicidal agent [80].

The studies of structure–activity relationship (SAR) of BTs have revealed that a change in the structure of the substituent group at the C2 position, in general, changes the bioactivity. Therefore, the biological profiles of this compound encourages researchers to design novel and efficient methods for the synthesis of BTs and their structural analogues. The development of these synthetic processes is undoubtedly one of the most significant challenges facing researchers.

Since the 2-substitutedBTs synthesized by A. W. Hofmann in 1887, organic chemists have developed synthetic processes to access BT compounds. Investigation into the synthesis of BT derivatives has been increasing over the years, and several reviews about this topic have been reported in the literature [81,82,83,84,85,86,87,88,89,90,91,92,93,94,95,96,96,97,98,99,100,101,102,103,104]. It is worth mentioning that at least eight reviews, performed in 2020 alone, have been found about the synthetic strategies and biological activities of BT nucleus derivatives [97,98,99,100,101,102,103,104]. In these works, we found that many protocols have been developed for the synthesis of BTs. However, only a few methods for the condensation of *o*-aminotiphenoles (ATPs) with carboxylic acids and their derivatives were discussed. Accordingly, in our article, we present a literature review on the aspects of the research progress related to the condensation of ATPs with carboxylic acids and their derivatives, such as acid chlorides, amides, esters, orthoesters, nitriles, and thioesters, including carbon dioxide (CO_2_), as starting materials. This report includes protocols ranging from those for conventional methods, such as the use of heterogeneous solid acid catalysts and reactions performed under mild and solvent-free conditions, to those for green chemistry methods, such as the use of ionic liquids, reactions under microwave irradiation, as well as the use of ultrasound energy performed in the last 20 years.

## 2. Results and Discussion

### 2.1. Condensation with Carboxylic Acids

Considering a retro-synthetically point of view, a carbon-sulfur (C2–S) bond of 2-substitutedBTs **5** can be cleaved to represent an *o*-amidethiophenol **3**, which can be obtained from the condensation of ATP **1** and a carboxylic acid **2** (Figure 1). This approach proceeds with the intramolecular cyclization of the *o*-amidethiophenol **3** via the SH group by an *o*-aryl-S cross-coupling protocol to the hydroxybenzothiazolidine ring **4**, which on heating or catalytic dehydration, produces the corresponding benzothiazole derivative **5** (Figure 1).

#### 2.1.1. Polyphosphoric Acid (PPA) as a Catalyst and a Dehydrating Agent

Polyphosphoric acid (PPA) has been extensively used as a good solvent for many organic compounds in organic synthesis. It has been used as one of the most effective reagents for acylation, alkylation, cyclization, and acid-catalyzed reactions. It is often the reagent of choice for a variety of synthetic transformations, such as dehydrations, rearrangements, and synthesis of nitrogen-containing heterocycles. PPA has also proved to be very useful in polymer synthesis.

Hein et al. used PPA as a solvent, a catalyst, and a dehydration agent in the condensation of ATP with carboxylic acid, ester, amide, or nitrile to generate 2-arylBTs and 2-alkylBTs in good yields. This procedure gives products not obtainable by other procedures [105].

The same authors used PPA in the condensation of substituted *o*-ATP with substituted 4-aminobenzoic acids at 220 °C to synthesize the 2-(*p*-aminophenyl)BTs **6a** (Figure 2) to be evaluated against breast cancer cell lines in vitro and in vivo [46]. ATP was condensed with substituted 4-aminobenzoic acids in PPA to give the 2-substitutedBTs **6b**–**d** (Figure 2). This route also afforded the 2-(2-aminopyridin-5-yl)BT **7** in a 59% yield from the reaction of 6-aminonicotinic acid and ATP in PPA (Figure 2) [106].

Rosenberg et al. used PPA to synthesize a polyBT **8** (Figure 3) by the condensation reaction of terephthalic acid (TA) with 4,6-diamino-1,3-dithiohydroxybenzene [107].

A mechanistic pathway for this condensation was proposed. ATP and benzoic acid react with PPA to produce *o*-ammoniumthiophenol **a** in a mixture of benzoic-phosphoric/benzoic-polyphosphoric anhydrides **b**, which react to form *o*-ammoniumphenylbenzoate **c** as an intermediate. This undergoes acyl migration to generate *o*-benzanilidethiophenol **d**, which undergoes an acid-catalyzed ring closure and then dehydration to form 2-phenylBT (Figure 4) [108].

Condensation of ATP with 4-acetylamino-3-hydroxybenzoic acid in PPA as a catalyst on heating at 150 °C for 12 h gave a direct route to access 2-(3-hydroxy-4-acetylamine)phenylbenzothiazole **9** (Figure 5) in a 61% yield [41].

ATP was condensed with 5-aminosalicylic acid on heating PPA at 200 °C for 8–10 h via cyclization to produce the 2-(2-hydroxy-5-aminophenyl)BT **10** (Figure 6) in good yield. 2-Azo-linked substituted BTs derived from **10** were synthesized to be evaluated for in vitro antibacterial activity against *Staphylococcus aureus* and *Escherichia Coli* strains by the Resazurin microtiter assay method (REMA) [109].

ATP was heated with 2-hydroxy-aminobenzoic acids in PPA at 220 °C to synthesize 2-(2-hydroxy, 4-aminophenyl)BT **11** and 2-(2-hydroxy, 5-aminophenyl)BT **12,** used as starting materials to synthesize positional isomers of BT-pyridone and BT-pyrazole containing disperse azo dyes, (Figure 7) [110].

Mathis et al. condensed 5-MeO-ATP with *p*-(methylamino)benzoic acid in PPA at 170 °C under N_2_ atmosphere for 1.5 h to generate the corresponding 2-(4-*N*-methylphenyl)-6-methoxyBT **13** in 21% yield, which was selectively O-demethylated at the 6-position to afford 6-OH-BT-1 **14** (Figure 8). Staining of the AD frontal cortex tissue sections with 6-OH-BT-1 indicated the selective binding of the compound to amyloid plaques and cerebrovascular amyloid [75].

Hutchinson et al. condensed ATP with first 4-amino-3-fluorobenzoic acid and then 4-amino-3-trifluoromethylbenzoic acid in PPA as a catalyst after heating at 110 °C for 0.5 h to generate 2-(4-amino-3-fluorophenyl)BT (Figure 4 (**15**) when R = F) and 2-(4-amino-3-trifluoromethylphenyl)BT (Figure 4 (**15**) when R = CF_3_) in 68% and 37% yields, respectively, to evaluate their in vitro antitumor biological properties [17b]. The lipophilic thioflavin-T analogue 2-(4-(methylamino)phenyl)BT **16** was obtained by heating ATP with *p*-N-methylaminobenzoic acid in PPA at 180 °C for 4 h in a 40% yield (Figure 4). This compound was reported to have a high affinity for amyloid plaque [111].

2-ChloromethylBT **17** (Figure 9) was obtained in 62% yield by cyclocondensation of ATP with chloroacetic acid on heating in the presence of PPA at 180 °C for 8 h [112].

Billeau et al. condensed ATP with several substituted benzoic acids by heating in the presence of PPA at 140 °C for 24 h to generate the corresponding 2-arylBTs **18a**–**v** and **19a**–**d** (Figure 5), which precipitated from water and were purified by recrystallization from CH**_2_**Cl**_2_** to give the products in 80–90% yields [113].

ATP and 2-picolinic acid were condensed in the presence of PPA while stirring at 120 °C under N**_2_** and then for 20 h at 160 °C. After separation and purification, the precipitate was recrystallized from methanol to afford 2-(*o*-pyridyl)BT **19a** (Figure 5) as a pale-yellow solid in a 45% yield (the table in Figure 5) [114]. Compound **19a** was used as a bidentate ligand to afford a palladium (II) complex. Antibacterial activity of the ligand and the corresponding complex against three Gram_−_negative and two Gram-positive microorganisms was performed, and the complex showed better microbial inhibition activity than the ligand and the palladium salt as reference.

Kini et al. refluxed ATP with substituted benzoic acids in the presence of PPA at a high temperature to get 2-arylBTs **20** (Figure 10) in 74% yield. Compound **20** was evaluated against the human cervical cancer cell lines as an anticancer drug [115].

Serdons et al. reported the condensation of ATP with *p*-nitrobenzoic acid in the presence of PPA at 150 °C for 1 h to give 2-(*p*-nitrophenyl)BT **21** in a 35% yield as the starting material for the synthesis of F-labeled 2-(*p*-fluorophenyl)BT **22** (Figure 11). The compounds were evaluated as amyloid imaging agents in Alzheimer’s disease [78].

An interesting method for the synthesis of naphthyridineBT derivatives **23** has been described by Kou et al. In this method, ATP undergoes cyclization on treatment with naphthyridine-3-carboxylic acids in the presence of PPA at 170–250 °C to afford compounds **23** in 10–60% yields (Figure 6) [116].

The condensation reactions of 4-amidino-substituted ATPs with 2,5-furan- and 2,5-thiophene dicarboxylic acid were performed on heating in PPA at 180 °C for 2 h. The bisamidinodiBTyl compounds **24** and **25** were isolated as hydrochloride salts in 35–76% yields (Figure 12) [117].

A single-step method to synthesize 2-arylBTs **26** in 50–60% yields has been reported from the condensation of ATP with substituted *p*-aminobenzoic acids in the presence of PPA on heating at 220 °C for 3 h (Figure 13) [118]. To improve the yield, 4-nitrobenzoic acid was condensed on heating at 180 °C for 5 h to afford 2-(*p*-nitrophenyl)BT in an 81% yield, with subsequent reduction using Fe/NH_4_Cl in refluxing 75% ethanol for 2 h, which gave access to 2(*p*-aminophenyl)BT **26a** (Figure 13). The same reaction conditions were used in the condensation of ATP with *p*-nitrobenzoic acid and 4-aminobenzoic acid to produce the corresponding 2-(*p*-aminophenyl)BT in a 90% yield (Figure 13) [119].

Isomeric-amidinoBTyl-disubstituted pyridines **27a**–**i** and pyrazine **28** were synthesized in 46–79% yields by the condensation reaction of pyridine and pyrazine dicarboxylic acids with 5-amidino- and 5-imidazolinyl-substituted ATPs. To improve yields, the quantity of the byproducts was reduced on heating in PPA first at 120–140 °C for 1 h and then for 2 h at 160–180 °C (Figure 7) [120].

In a method developed by Santos et al., ATP was condensed with amino acids, such as glycine and *d*-valine, as hydrochlorides on heating at 220 °C in the presence of PPA for 4 h to give the 2-substitutedBTs **29a** and **29b** in low yields (Figure 14). However, 53.2% and 41.0% yields of the respective compounds were obtained when the corresponding amino acid ethyl esters was used [121].

Condensation of 5-amidino-substituted ATP with carboxylic acids by heating in the presence of PPA at 110–180 °C for 3 h as a method to prepare 6-amidino-2-aryl/heteroarylBTs **30a**–**u** in 31–74% yields was also reported (Figure 8) [122].

#### 2.1.2. Other Acids as Catalysts

The use of PPA requires high temperatures (110–220 °C) and long reaction times (1–12 h). These harsh conditions depend on the starting materials’ stability and are a limitant to generalizing this condensation. Therefore, as an alternative to PPA, other acids as catalysts have been designed to be used in this condensation.

2-SubstitutedBTs **31** were obtained from ATP and the corresponding carboxylic acid by treatment with P_2_O_5_/MeSO_3_H (1/10, *w*/*w*) while warming. The reaction was effective for a wide range of aliphatic and aromatic carboxylic acids. The general procedure involves treating a mixture of P_2_O_5_/MeSO_3_H (1/10, *w*/*w*) and *o*-ATP in the ratio 1.5 g/1.0 mmol with 1.0 equivalent of the corresponding carboxylic acid and warming for 10 h [123].

Yildiz et al. reported the condensation of ATP with several carboxylic acids in the presence of trimethylsilylpolyphosphate ester (PPSE) as the cyclodehydration reagent and on refluxing for 3–4 h (140 °C) to afford a series of 2-substitutedBTs **32** in 43–72% yields (Figure 15). All 2-substitutedBTs were tested for their antibacterial activities against Gram-positive and Gram-negative bacteria and antifungal activity against the fungus *Candida albicans* [124].

Ge et al., in a one-pot synthesis, condensed ATP with trifluoroacetic acid or difluoroacetic acid in the presence of PPh_3_, excess of Et_3_N, and CCl_4_ on refluxing to produce 2-trifluoro- and 2-difluoromethylBTs **33** (Figure 16) in 72% and 85% yields, respectively (Figure 16). The proposed mechanistic procedure is based on the electrophilicity of the imino group of the fluorinated *N*-aryl imidoyl chloride intermediate **b** (Figure 16). The (-SH) nucleophilic group at the *ortho* position on benzene produces an intramolecular nucleophilic substitution and leads to the formation of 2-fluoroalkyl BTs [125].

Jiang et al. used triphenylphosphine (Ph_3_P) as the catalyst in the condensation reaction of ATP with bromodifluoroacetic acid in the presence of 3 molar equivalents of carbon tetrabromide (CBr_4_) in refluxing toluene for 24 h to prepare 2-bromodifluoromethylBT **34** in a 65% yield to be used as a building block in the subsequent reactions. The reaction mechanism involves the formation of imidoyl bromide and intramolecular ring-closure reaction, similar to the mechanism represented in Figure 16 [126].

Sharghi et al., in a one-pot procedure, condensed ATP with several aliphatic or aromatic carboxylic acids using a MeSO_3_H/SiO_2_ system as a dehydrating catalyst at and heating at 140 °C for 2–12 h to synthesize 2-substituted aliphatic and aromatic BTs **35** in 70–92% yields (Figure 17). A simple work-up, using diverse carboxylic acids, and easy handling reaction conditions are the benefits of this method. The 2-(*o*-, *m*-, and *p*-NO_2_-phenyl)BTs were obtained in 35%, 11%, and 53% yields, respectively [127].

For instance, 2-chloromethylBT was obtained in a 92% yield on the condensation of ATP with chloroacetic acid and heating the product at 140 °C for 2.5 h, using MeSO_3_H/SiO_2_ as the catalyst, in the synthesis of BT derivatives [128].

The cyclocondensation of ATP and aromatic carboxylic acids to give 2-substitutedBTs **36** in 60–87% yields has been carried out under mild conditions using tetrabutylammonium bromide (TBAB) as the reaction medium and triphenyl phosphite (TPP) as the catalyst (Figure 18). Shorter reaction times, rapid isolation of products, and good yields are advantages of this method. The reaction was found to be general and quite tolerant to the nature of the substituted carboxylic acids. The mechanism shown in Figure 19 is assumed to operate. Triphenyl phosphite reacts with aromatic carboxylic acids to produce the corresponding ester **a**, which reacts with o-aminothiophenol to form the phosphoester intermediate **b**. The thiol group of phosphoester **b** attacks the C=O to give, after intramolecular cyclization, the title products [129].

N-Boc-protected valine amino acid reacts with ATP in the presence of EDC, HOBT, and DCM, followed by the intramolecular Mitsunobu reactions in the presence of DIAD, PPh_3_, and TEA to afford 2-substitutedBT **37** as an intermediate (Figure 20) [130].

Fifteen 5-substituted-2-(pyridyl)BTs **38** and **39** were synthesized in 34–42% yields by the condensation reaction of substituted ATPs with the corresponding pyridinylcarboxylic acids under the PPA/H_3_PO_4_ system (Figure 21). Activities of these synthesized compounds were evaluated on Bcap-37, HCT-15, and HepG2 tumor cells in vitro by a standard MTT assay. 5-Fluorouracil (5-FU) was used as the positive control [131].

Recently, condensation of ATP with piperidine-4-carboxylic acid on heating PPA has been carried out to synthesize 2-(piperidin-4-yl)BT **40** as an intermediate to be used as a building block (Figure 22) [132].

A series of 2-(3-butynoicamidophenyl)BTs **41** were synthesized starting from 4-fluoro-3-nitrobenzoic acid and ATP. Their antitumor activities against human tumor cell lines (HCT116, Mia-PaCa2, U87-MG, A549, NCI-H1975) were evaluated by an MTT assay [133].

2-SubstitutedBTs **42** were synthesized in 72–92% yields from the reaction of ATP with a set of substituted aromatic carboxylic acids using Samarium(III) triflate as a catalyst (Figure 23). Short reaction times, aqueous reaction media, and easy work-up are the advantages of this method. The catalysts were reused without any appreciable loss of efficiency [134].

A simple trituration method for the synthesis of 2-substitutedBTs **43** from the reaction of N-protected amino acids and ATP using molecular iodine as a mild Lewis acid catalyst has been proposed (Figure 24). The reaction occurs in one step that lasts for 20–25 min in solvent-free conditions to afford the products in 66–97% yields [135].

#### 2.1.3. Condensation on Direct Heating

In addition to the harsh conditions, such as toxic solvents, strong acidic conditions, high temperatures, and long reaction times, employed in condensation reactions of ATPs with carboxylic acids, the use of PPA as a catalyst and the side reactions carried out lead to the lowering of selectivities and low yields. Therefore, there is a strong demand for a highly efficient and environmentally benign method for the synthesis of these heterocycles. In this sense, the use of ionic liquids (ILs) as green solvents in organic synthesis has gained considerable importance due to their solvating ability, negligible vapor pressure, and easy recyclability. In addition, they have been shown to promote and catalyze organic transformations due to their high polarity. ILs can also be recovered and recycled many times with marginal loss.

Substituted 2-arylBTs **44** can be rapidly (10 min) synthesized in 80–94% yields by the condensation of a set of carboxylic acids with ATP under atmospheric conditions using the ionic liquid 1-butyl-3-methyl-imidazoliumtetraflouroborate ((bmim)BF_4_) as a dissolvent and dehydrating agent at 100 °C (Figure 25) [136].

In an autocatalytic procedure, ATP and trifluoroacetic acid (TFA) were heated to 70 °C for 16 h to produce the corresponding 2-(trifluoromethyl)BTs **45a** (Figure 26). Evaporation of the excess TFA afforded the product in a 99% yield. However, the same reaction with the electron-deficient 4-trifluoromethyl-substituted *o*-aminothiophenol produced the corresponding 2-trifluoromethylBT **45b** in only 58% yield [137].

A metal-free methodology by the use of a base as an oxidant has been developed for the condensation of ATP with oxalic and malonic acids followed by a decarboxylation to afford BT **46a** and 2-methylBT **46b** in 80% and 82% yields, respectively (Figure 27). Easy work-up procedure, high yield, and easy isolation of products are the advantages of this methodology [138].

### 2.2. Condensation with Esters

The condensation of esters with ATPs produces the corresponding thioester **a** (Figure 4). The acyl group migrates to form the amide **b** (Figure 4), which is cyclicized and then dehydrated with the aid of a Brönsted or Lewis acid to the corresponding 2-substitutedBTs, as depicted in Figure 4.

Substituted ATPs have been condensed with methyl 4-amino-3-iodobenzoate ester under PPA on heating to 220 °C to produce 2-(4-aminophenyl)BT **47** in a 14% yield, instead of the expected iodinated compound (Figure 28) [106]. In acid conditions, the iodo group is electrophylically substituted. The ATP amino group is cauterized and the nucleophile is blocked to directly form the corresponding amide and the thioester. Then the acyl group migration is carried out at a high temperature.

Khalil et al. condensed the amino ethyl ester 5-carbonyl-(3,4-diamino-2-ethylcarboxylate thieno[2,3-*b*]thiophene-5-yl)-8-hydroxyquinolin with ATP by heating in the presence of PPA at 160 °C for 3 h and then carried out neutralization with aqueous ammonia to afford the corresponding 2-substitutedBT **48** in an 80% yield (Figure 29) [139]. In this case, the neutralization regenerates the ATP amino group to form the corresponding amide for the cyclization to be continued.

Phenolic esters were efficiently converted to 2-substitutedBTs **49** in a one-pot reaction by treatment with ATP in the presence of a catalytic amount of K_2_CO_3_ and then converted to a thiolate salt on heating with *N*-methyl-2-pyrrolidone (NMP) at 100 °C (Figure 30). The formation of a thioester **b** with the elimination of the corresponding phenol and migration of the acyl group to the nitrogen atom **c** can be proposed, followed by dehydration [79].

Polymer-bound esters were treated with ATP in the presence of AlMe_3_ as the Lewis acid dehydrating agent in refluxing toluene to afford the 2-substitutedBts **50** as the cleavage products in a 46–75% yields (Figure 31) [140].

Shantakumar et al. synthesized 2-(ethylcarboxilate)BT **51** in a 53% yield by the condensation of ATP and diethyl oxalate in a mild reflux for 4 h. During heating, the temperature decreased from 147 to 93 °C [63]. The same condensation has been carried out on refluxing for 8 h to afford compound **51** in a 92% yield (Figure 32) [141].

Sheng et al. condensed ATP with ethyl chloroacetate in dichloromethane as a dissolvent on heating from 0 °C to room temperature to afford 2-chloromethylBT **17** in a 68.7% yield (Figure 33) [142].

Santos et al. condensed ATP with glycine and *d*-valine ethyl ester hydrochlorides in the presence of PPA, heating at 220 °C for 4 h, to give the corresponding 2-substitutedBTs **29a** and 29**b** in 52.3% and 41% yields, respectively (Figure 34) [121].

5-Fluoro ATP potassium salt was reacted with HCl, followed by the addition of (*R*)-4-methyl-oxazolidine-2,5-dione derived from *d*-alanine, to give the (1*R*)-1-(6-fluoro-1,3-BT-2-yl)ethanamine hydrochloride **52**, which was transformed to its *p*-toluenesulfonate salt in an 81% yield (Figure 35). This compound was used as the starting material in the synthesis of a series of diamides [143].

### 2.3. Condensation with Orthoesters

The general mechanistic pathway of condensation reactions of ATPs with orthoesters has been proposed as represented in Figure 36.

The condensation of ATP with three ethyl orthoesters was carried out in the presence of catalytic quantities of H_2_SO_4_ on refluxing to produce the corresponding 2-substitutedBTs **46** in a 75–85% yield (Figure 37) [144].

In an efficient procedure, 2-chloro-1,1,1-triethoxyethane was condensed with ATP in a versatile synthesis of 2-chloromethylBT **17 [145]**.

A series of 2-alkylBTs (R = H, Me, Et) **46a**–**c** were synthesized in 81–86% yields from the reactions of ATP with orthoesters in the presence of catalytic amounts of Bi(III) salts, such as Bi(TFA)_3_, Bi(OTf)_3_, and BiOClO_4_xH_2_O, under solvent-free conditions at room temperature. High conversion, very short reaction times, cleaner reaction profiles, solvent-free conditions, straightforward procedure, and use of low-toxic catalysts are the features of this protocol [146]. The same condensation was carried out in the presence of catalytic amounts of the eco-friendly and inexpensive ZrOCl_2_·8H_2_O under solvent-free conditions to afford the compounds **46a**–**c** in 93–97% yields. The reusability of the catalyst, high yields, very short reaction times (4–6 min), solvent-free reaction conditions, and an easy experimental and work-up procedure are the advantages of this protocol [147].

Silica-supported fluoroboric acid HBF_4_-SiO_2_ was used as a catalyst in the condensation of ATP with orthoesters under solvent-free conditions at room temperature. A simple and environmentally benign synthesis of 2-aliphaticBTs **46a**–**e** in 84–97% yields was carried out by this procedure in 45 min (Figure 38) [148].

Bastug et al. condensed substituted ATP with functionalized orthoesters under mild conditions to prepare various 2-substitutedBTs **53** in 90–95% yields. The condensations were carried out at room temperature in the presence of 1 equiv. of BF_3_ OEt_2_ and were complete within 3 h. Multifunctional heterocycles are the specialty of this protocol (Figure 39) [149].

*o*-Benzenedisulfonimide was used as an efficient catalyst in the condensation of ATP with various orthoesters, giving the respective 2-substitutedBTs **54** in 1–2 h in 86–92% yields. The reaction conditions were very simple. The catalyst was easily recovered and reused (Figure 40) [150].

An eco-friendly procedure for the condensation of ATP with orthoesters employed 15% mol of catalytic ammonium chloride (NH_4_Cl) in H_2_O or ethanol to produce 2-substitutedBTs **46** in a short reaction time (30 min), involving easy work-up, and in a high yield (87–93%) (Figure 41) [151].

A solvent-free method, with sulfonated rice husk ash (RHA) as the solid acid catalyst, was developed for the condensation of ATP with orthoesters in the preparation of BT **46a** and 2-methylBT **46b** in 92% and 95% yields in 1 min. The catalyst could be reused five times without any loss of its catalytic activity (Figure 42) [152].

### 2.4. Condensation with Acyl Chlorides

In this method, acid chlorides react with ATPs to produce directly the corresponding amides as intermediates. In general, tertiary amines are used to trap the chlorohydric acid produced in the reaction. After the cyclization, the generated ammonium chloride salt formed is used as a catalyst in the dehydration of the 2-hydroxy thiazolidine intermediate to produce 2-substitutedBTs. Sometimes, any base is used. In this case, hydrochloric acid is used as the dehydrating agent.

Via a general method, ATP with 1-methyl-pyrrolidinone as the solvent was condensed with a stoichiometric amount of the corresponding acid chloride, added slowly at room temperature under an inert atmosphere. The corresponding 2-arylBTs **55** were obtained in 82–95% yields (Figure 43). The reaction mixture was heated at 100 °C for 1 h. The intermediate di-irido and the six-coordinated mononuclear iridium (III) dopants of the above ligands were synthesized and characterized [153,154].

4-Methyl-ATP was condensed with chloroacetyl chloride in the presence of Et_3_N and EtOAc as the solvent on stirring from 0 °C to room temperature to produce in situ the corresponding 5-methyl-2-chloromethylBT **56**, which was treated with thiourea to afford the more stable isothiouronium salt (Figure 44) [155].

BTyl compounds **59a** and **59b**, synthesized from the corresponding dialdehydes **57a**,**b** and 4-amino-3-mercaptobenzonitrile, were converted to the corresponding chlorocarbonyl derivatives **61a** and **61b**, which were condensed with 4-amino-3-mercaptobenzonitrile by Route A to produce the biscyanoBTyl compounds **61a** and **61b** in about 75% yields (Figure 45). By Route B, 5-(4-carboxyphenyl)-2-furylcarboxylic acid **58a** and 5-(4-carboxyphenyl)-2-thienylcarboxylic acid **58b** were converted to the dichlorocarbonyl derivatives **60a** and **60b** and then were condensed with 4-amino-3-mercaptobenzonitrile to afford **62a** and **62b** in about 35% yields (Figure 45). Route B has one step less than Route A. However, the overall yield of the reactions was considerably lower [156].

A regioselective one-pot synthesis of 2-arylBTs **63** was achieved in excellent isolated yields by the condensation of ATP and substituted benzoyl chlorides under ambient conditions using the ionic liquid mixtures 1-butylimidazolium tetraflouroborate ([Hbim]BF_4_) and 1,3-di-*n*-butylimidazolium tetrafluoroborate ([bbim]BF_4_) as the reaction medium and the reaction promoter, respectively (Figure 46). The absence of a catalyst, room temperature conditions, and non-volatile ILs makes this protocol green and environment friendly [157].

Karlsson et al. carried out the same condensation reaction of ATP with 4-nitrobenzoyl chloride by applying *N*-methyl-2-pyrrolidone (NMP) as an oxidant and heating at 100 °C for 1 h to give 2-(*p*-nitrophenyl)BT [158].

Chen et al. condensed ATP with isophthaloyl or terephthaloyl fluorides attached to a polyethylene glycol methyl ether PEG resin in a liquid-phase synthesis of two 2-aromaticBTs **64** (Figure 47). The cleavage from PEG was achieved by treatment with sodium methoxide in methanol for 12 h. The compounds **64** were obtained in four steps, with poor isolated yields (22% and 37%) [159].

Kondaskar et al., in a one-pot reaction, generated acyl chlorides from carboxylic acids and condensed these with ATP under acid- and catalyst-free conditions to generate 2-alkyl and 2-arylBTs **65** (Figure 48) [160].

Wu et al. condensed the zinc salt of 4-amino-3-mercaptobenzoic acid with *p*-nitro benzoyl chloride by heating in the presence of pyridine at 80 °C for 1 h to give the intermediate 2-(4-nitro-phenyl)-BT-6-carboxylic acid **66** in an 83% yield (Figure 49). This was treated with SOCl_2_ to be converted into the acyl chloride **67**, followed by coupling with 5-substituted ATPs in refluxing chlorobenzene under heat treatment for 3 h to produce the corresponding 2-(4′-nitrophenyl)-(6-BTyl)BTs **68** in 73–79% yields [161].

In 2010, Racané et al. reported the condensation of cyano- or nitro-substituted ATPate with 4-cyano- and 4-nitrobenzoylchloride under reflux in acetic acid (AcOH) for 4 h to prepare the corresponding *bis*-nitrile and nitro-nitrile 2-phenylBTs **69** in 71–84% yields (Figure 50). Amidino- or imidazo-substituted 2-phenylBTs were prepared by the corresponding precursors in zwitterionic form with 4-nitrobenzoylchloride in a good yield (70%). 2-(*p*-Nitrophenyl)BTs were used as starting compounds for the preparation of 2(*p*-amidino-phenyl)BTs by the Pinner reaction. All compounds except diamidino-substituted 2-phenylBT showed exceptionally prominent tumor-cell-growth inhibitory activity [162].

The condensation reaction of 2,5-furan and 2,5-thiophene di-acid chlorides with 5-amidino-substituted ATP was performed in refluxing glacial acetic acid for 4 h (Figure 51). The bisamidino diBTyl compounds **70a**,**b** and **71a**,**b** were isolated as hydrochloride salts in good yields, of about 72–88% [117].

In simple reaction conditions, the condensation of ATP with various acid chlorides in the presence of *o*-benzenedisulfonimide as the catalyst has been carried out in 2–3.5 h to give the respective 2-substitutedBTs **72** in 42–87% yields (Figure 52) [150].

The intermediate BT alkylating agents **73** were prepared in 55–76% yields from the condensation reaction of ATP with chloroacyl chlorides while stirring in toluene for 15 min at room temperature (Figure 53) [163].

Pyridine-2,6-dicarbonyl dichloride was condensed with the switter ionic amidino-substituted ATPs in refluxing acetic acid for 3 h to give the dihydrochloride salts **74a** and **74b** in 78% and 75% yields, respectively (Figure 54) [120].

The same authors condensed isophthaloyl and terephthaloyl dichlorides with the corresponding 5-amidinium-ATPate and 5-(imidazolinium-2-yl)-*o*-ATPate switter ions in refluxing acetic acid for 3 h to give the corresponding *m*- and *p*-phenylene-bisBTs **75a**–**e**, which were isolated as dihydrochloride salts in a 90% yield (Figure 9) [164].

Kumar et al. condensed ATP with benzoyl chloride and *o*-chlorobenzoyl chloride using an efficient and green catalyst NaHSO_4_-SiO_2_ to promote the solvent-free synthesis of 2-phenylBT **76a** in 89% and 2-(*o*-chlorophenyl)BT **76b** in 86% yields. The best results were obtained on refluxing at 100 °C for 12 h without a solvent. The catalyst could be easily recovered [165].

The synthesis of S-alkyl/arylBT-2-carbothioates **77** through a three-component reaction was reported from the condensation of substituted ATPs with thiols/oxalyl chloride system using 10 mol % of tetrabutylammonium iodide (TBAI) as the catalyst in acetonitrile as a solvent (Figure 55). BTs and thioesters were formed via simultaneous C–N and C–S bond formation in 56–80% yields with several substrates. The synthesized derivatives were tested for their antimicrobial activity against the protozoan parasite *Leishmania donovani*, a causative agent of visceral leishmaniasis (VL) [166].

To understand the reaction mechanism, treatment of ATP with oxalyl chloride was carried out in the same conditions; however, the corresponding bis-BT **78** was obtained in only a 25% yield (Figure 56).

The formation of the S-alkyl BT-2-carbothioates **77** can be proposed, as TBAI reacts with thiol to generate thiolate anion **a**, which attacks a side of oxalyl chloride to form a thioester **c** as an intermediate (Figure 57). Finally, ATPate anion **a**, obtained from TBAI and *o*-ATP, undergoes condensation with the in situ formed thioester **c** to give the products (Figure 57).

A useful protocol for the preparation of substituted 2-aminoBTs **78** was presented. Substituted ATPs were reacted with thiocarbamoyl chlorides under the catalysis of copper in a tandem manner to produce compounds **78** with 70–91% yields (Figure 58). The broad substrate scope, a short reaction time, mild reaction conditions, easy performance, and excellent yields make this protocol attractive for the preparation of some biologically active compounds [167].

Various carboxylic acid chlorides were reacted with ATP in the presence of KF/Al_2_O_3_ (25% mol) as a heterogeneous base catalyst in refluxing dry acetonitrile at room temperature to form the corresponding 2-substitutedBTs **79** in 87–97% yields, with good recyclability of the catalyst (Figure 59). However, the condensation of o-ATP with the mixture of acetic and benzoic anhydrides as carboxylic acid derivatives produced the BTs in 90% and 91% yields [168].

### 2.5. Condensation with Nitriles

A proposed mechanism for this reaction is via the formation of an amidine **a** as an intermediate and subsequent cyclization followed by a deamination to afford the 2-substitutedBT (Figure 60). This reaction can be catalyzed to accelerate the reaction.

The 2-(4-amino-3-chloro)BT **80a** and 2-(4-amino-3-chloro-5-methyl)BT **80b** were prepared from the condensation of ATP with 4-amino-3-chlorobenzonitrile and 4-amino-3-chloro-5-methylbenzonitrile, respectively, on heating in PPA at 200 °C (Figure 61) [106].

Mokhir et al. condensed ATP and malonodinitrile in the presence of glacial acetic acid and ethanol as a solvent. After stirring for 1 h, 2-cyanomethylBT **81** precipitated as yellow crystals (Figure 62) [169].

Zandt et al. reported the synthesis of 4-fluoro-2-hydroxy-*N*(4,5,7,-trifluoroBT-2-yl-methyl)-benzamide **82** using *N*-cyanomethyl-4-fluoro-2-hydroxy-benzamide and *o*-amino-4,5,7-trifluorothiophenol hydrochloride in refluxing ethanol for 24 h (Figure 63) [170].

Manfroni et al. generated in situ substituted ATPs to be immediately condensed with ethyl cyanoacetate on heating at 120 °C under stirring and N_2_ flux. After 15 h, the 5-substituted ethyl (BT-2-yl) acetates **83** were isolated, **83a** and **83b** in 22% and 14% yields, respectively, **83c** and **83d** in 93% and 74% yields, respectively (Figure 64) [171].

Sun et al. used 10 mol % of copper acetate Cu(OAc)_2_ as a catalyst in the presence of Et_3_N in the condensation of substituted ATPs with several nitriles on heating in ethanol at 70 °C for 6 h to afford 2-substitutedBTs **84** in 75–97% yields (Figure 65). According to the proposed reaction mechanism, the formation of a sulfilimine **b** and intramolecular cyclization occur to afford compounds **84** [172].

A convenient and efficient method for the synthesis of 2-substitutedBTs **85** in 50–95% yields was carried out from the condensation of substituted ATPs with substituted benzonitriles by the use of trifluoromethanesulfonic acid on heating at 100 °C for 12 h (Figure 10). The Bronsted-acid-catalyzed cyclization reaction was performed under metal- and solvent-free conditions [173].

### 2.6. Condensation with Amides

In this case, deamination occurs to form the corresponding amide. Then, cyclization and dehydration produce 2-substitutedBTs. In some cases, this reaction requires to be catalyzed.

3-(BT-2-yl)coumarins **86** were prepared in 61–70% yields from the condensation of ATP with the corresponding 2-iminocoumarin-3-carboxamide in reflux with the minimum amount of *n*-butanol until the evolution of ammonia stops (1–1.5 h) (Figure 66) [174].

The synthesis of several substituted BTs **87** in 50–94% yields was carried out from the condensation of substituted ATPs with 2-acylpyridazin-3(2H)-ones as acyl transfer agents under transition-metal-free and eco-friendly conditions. The reaction was efficient, green, and economical, with several applications in organic synthesis and in medicinal and industrial chemistry (Figure 67) [175].

An efficient and convenient one-pot procedure was developed for the synthesis of a series of substituted BTs **88** in up to 95% yields. This protocol uses various substituted ATPs and N-substituted formamides in zinc-catalyzed cyclization in the presence of poly(methylhydrosiloxane) PMHS (Figure 68) [176].

In an efficient and mild protocol, several substituted ATPs were condensed with thioamides to produce 2-substitutedBTs **89** in 62–93% yields using CBr_4_ as the catalyst under solvent- and metal-free conditions (Figure 69). This condensation process involves the activation of a thioamide through halogen bond formation between the sulfur atom of the thioamide and the bromine atom of the CBr_4_ molecule. The presence of halogen-bonding interaction between *N*-methylthioamides and tetrabromomethane was demonstrated. Substituted 2-alkyl- and 2-arylBTs could be obtained from this methodology [177].

A simple, economical, and metal-free approach for the synthesis of 2-substitutedBTs **90** was reported in 60–87% yields from the condensation of substituted ATP and DMF derivatives, using imidazolium chloride (50% mmol) as the only promoter, without any other additive (Figure 70) [178]. The proposed mechanistic pathway is as described in Figure 71.

### 2.7. Condensation under Microwave Irradiation (MWI)

The Montmorillonite KSF Clay minerals act as strong Bronsted acids and have been used as solid, non-corrosive catalysts in the condensation of two orthoesters with ATP under toluene reflux or a solvent-free condition under MWI to produce BT and 2-methylBT **46** in 69% and 74% yields, respectively (Figure 72) [179].

Reddy et al. synthesized 2-(1,1,1-trifluoroacetonyl)BT **91** in a 93% yield by condensation of ATP with trifluoroacetyl ketene diethyl acetal under MWI in toluene for 8 min (Figure 73) [180].

Chakraborti et al., in a direct condensation of ATP with aromatic or aliphatic carboxylic acids under MWI in the absence of a solvent, produced the corresponding 2-substituedBTs **92** in 45–97% yields (Figure 74) [181].

To improve yields, a MWI-assisted procedure to condense ATP with 2-chloroacetyl chloride in acetic acid to produce 2-chloromethylBT **17** was carried out for a 90% yield in 10 min [182].

Several aliphatic and aromatic carboxylic acids were condensed with ATP in the presence of 0.35 equivalents of Lawesson’s reagent as promoters under solvent-free MWI (300 W, 190 °C) for 0.5–4 min to afford 2-substitutedBTs **93** with good yields (Figure 75). In this methodology, Lawesson’s reagent in situ converts carboxylic acids to thiocarboxylic acids, which are cyclocondensed with desulfhydration. The same protocol was used directly with thiobenzoic acid to afford 2-phenylBT in quantitative yields in 1 min [183].

Gupta et al. condensed ATP with various benzoic acid derivatives employing a catalytic quantity of molecular iodine (I_2_) as the dehydrating agent in a one-pot, solid-phase, solvent free, and MWI-assisted reaction to obtain 2-arylBTs **94** in 10 min with 60–70% yields (Figure 76) [184]. Less time consumption and lower cost compared with the use of PPA and 1-pentyl-3-methyl imidizolium bromide [pmim]Br catalyst are the benefits of this procedure.

MWI-assisted condensation of four resin-bound esters with three substituted ATPs (4 equiv.) in 15% of methane sulfonic acid/1,2-dichlorobenzene system was carried out to give twelve 2-substitutedBTs **95** in 68–93% yields (Figure 77) [185].

Rauf et al., in an efficient and solvent-free one-pot method, condensed ATPs with various fatty acids with or without P_4_S**_10_** as the catalyst under MWI (3–4 min) to afford 2-substitutedBTs **96** (Figure 78) [147]. When P_4_S**_10_** was used as the catalyst, the power and the reaction time were lowered from 100% to 60% and from 28–30 to 3–4 min, respectively, with an increase in the yields from 60–65% to 80–85% [186].

Khoobi et al. prepared 3-(2-BTyl)coumarin derivatives **97** in a 88% yield via the reaction of 3-cyanocoumarin with ATP in the presence of a catalytic amount of acetic acid (AcOH) under reflux overnight (Figure 79) [187]. The same reaction with a series of substituted *o*-aminothiophenoles and substituted 3-cyanocumarines was carried out by the use of acetic acid or catalytic amounts of H_6_[PMo_9_V_3_O_40_] (HPMo) in a MWI-assisted protocol in 15 min to synthesize compounds **97** with 62–88% yields [188].

In a one-pot procedure, propylphosphonic anhydride (**^®^**T3P) promoted the cyclo-condensation of ATP with aromatic and aliphatic carboxylic acids for the quick preparation (10 min) of 2-substitutedBTs **98** in 78–97% yields, under MWI, (Figure 80) [189].

The same procedure was used to prepare 2-(*para*-pyridyl)BT (*p*-PBT) and 2-(*para*-pyridyl)BT N-oxide (2-(N-O *p*-PBT). The synthesized PBTs were used as ligands to react with metal ions such as Co(II), UO_2_(II), Sm(III), Eu(III), and Zn(II)) to afford a series of metallic complexes [190]. The photoluminescence properties of Samarium and Europium complexes were studied.

Panda et al. discovered that an excess of ATP (10 equiv.) and benzoyl benzotriazolide (1 equiv.) was coupled under microwave heating at 70 °C for 1 h in 91–95% yields without any catalyst or solvent used to produce 2-substituteBTs **99** (Figure 81). The liquid *o*-aminothiophenol acts as both a solvent and a reagent. The same reaction under conventional heating was less efficient [191].

The same “suspension-in-water” protocol was used in the coupling of ATP with peptidylbenzotriazolides to afford 2-peptidylBTs **100a**–**o** in 70–89% yields (Figure 82).

Condensation of ATP with (un)substituted *p*-aminobenzoic acids under the action of melamine formaldehyde resin (MFR)-supported sulfuric acid under microwave irradiation (900 W) and solvent-free conditions was performed (Figure 83). The simple method afforded 2-(4-aminophenyl)BTs **101** in 81–88% yields in 6–8 min. The catalyst could be recycled three times, with the yield lowering from 87% to 80% [192].

A metal-free and catalytic method for the synthesis of 2-substitutedBTs **102** and **103** in 815–92% yields was carried out involving the condensation of ATP with *N*-protected amino acids and carboxylic acids under MW irradiation (Figure 84). The reactions proceeded through a two-step mechanism. The coupling step was carried out with ethyl 2-cyano-2-(2-nitrobenzenesulfonyloxyimino)acetate (I) (*ortho*-No-sylOXY), and the cyclization step took place with the use of *para*-toluenesulfonic acid as the catalyst [193].

Luo et al. found that 2-chloromethylBT **17** could be obtained in an 87% yield from the condensation of ATP with chloroacetyl chloride in acetic acid under MW irradiation for 10 min. The MW-assisted procedures were efficient and environmentally friendly, and used less time, compared with traditional methods [194].

A method for the synthesis of a series of 2-substitutedBTs **104** in 82–92% yields was developed involving the condensation of ATP with a series of carboxylic acids using Amberlyst-15 as a recyclable catalyst under ultrasound irradiation in water (Figure 85). This methodology does not use hazardous organic solvents and is carried out in an inert or anhydrous atmosphere [195].

### 2.8. Condensation of o-Aminothiophenol Disulfides (ATPDs)

In this condensation reaction, a reducing agent was required before or after the amide formation to afford the corresponding amidethiophenoles, which were cyclized to the 2-substitutedBTs.

Interaction of the bis(2-amino-4-fluorophenyl)disulfide with either 3-methyl-4-nitrobenzoyl chloride or 4-nitrobenzoyl chloride gave the corresponding amides, which were cyclized and reduced with tin(II) chloride dihydrate and HCl in ethanol, under reflux for 2 h, to afford the pure 5-fluoro-2-(4-aminophenyl)BTs **105** (Figure 86). 2-(4-Amino-3-methylphenyl)-5-fluoroBT was resistant to metabolic C-hydroxylation and because of its potency and broad spectrum in vitro, this compound was proposed as a clinical candidate [25].

2-PhenylBTs **106** were cleanly and efficiently synthesized from reductive cyclization of the disulfide of ATP with a set of acyl chlorides, employing different reducing agents [196].

Shi et al., in short reaction times, synthesized 2-arylBTs **107** in 70–92% yields via the reductive cyclization of bis-(2-benzalaminophenyl)disulfide promoted by a titanium tetrachloride (TiCl_4_)/Samarium (Sm) system using tetrahydrofuran (THF) as a solvent (Figure 87) [197].

2-ArylBTs **108** were synthesized using a PCl_3_ and DBU as the organic base to promote the cleavage/acylation/cyclization tandem reaction of the disulfides of ATP and aromatic carboxylic acids. PCl_3_ acted as both acylating and disulfide cleaving reagents, which prompted the disulfides of ATP to react with the aromatic carboxylic acid (Figure 88) [198].

Coelho et al. described a methodology to condensate ATP disulfide with 4-substituted benzoic acids in the presence of tributylphosphine as both the promoter of disulfide bond cleavage and the activating agent for coupling with carboxylic acids containing donor/withdrawing substituents for preparing 2-arylBTs **109** in 47–92% yields (Figure 89) [199].

This method was used to prepare the amyloid probe 2-(4-aminophenyl)-6-methoxyBT. A plausible mechanism was proposed as depicted in Figure 90. A nucleophilic attack of tributylphosphine promotes the cleavage of the S–S bond of disulfide to produce a thiolate phosphonium salt **a**. Thiolate deprotonates the carboxylic acid, and the generated carboxylate reacts with the phosphonium salt to form a pentacoordinate acyloxyphosphonium intermediate **b**. Intramolecular nucleophilic attack of the amino group on the carbonyl produces amide **c** and tributylphosphoxide. Finally, a nucleophilic attack of the thiol group on the carbonyl of the amide **c** followed by dehydration to leads to the BT.

### 2.9. Condensation by Cyclization of CO_2_ as the Raw Material

In this reaction, the CO_2_ must be activated to form a formiate ester or formamide as an intermediate to be condensed with ATPs to produce the corresponding BTs.

A series of substituted *o*-ATPs were cyclized with CO_2_ in the presence of diethylsilane and 1,5-diazabicyclo[4.3.0]non-5-ene (DBN) as the catalyst at 5 MPa to synthesize substituted BTs **110** in 35–82% yields (Figure 91) [200]. The proposed mechanism shows that hydrosilane played an important role in activating CO_2_ in the formation of compounds **110**.

The same authors cyclized substituted ATPs with CO_2_ and hydrosilane to produce a series of substituted BTs **111a**–**j** in up to 99% yields under metal-free and mild conditions using the acetate-based ionic liquid ([Bmim][OAc]) as a catalyst at 60 °C and 0.5 Mpa (Figure 92) [201]. The IL was reused five times without change. A possible mechanism was proposed as shown in Figure 88. CO_2_ was activated by the IL to produce intermediate **a**. The Si–H bond of (EtO)_3_SiH **b** was activated by the IL to facilitate the insertion of **a** to form formoxysilane intermediate **c**; meanwhile, *o*-ATP was activated via the hydrogen bond by the anion [**^−^**OAc] of IL and the nucleophilic N atom attacked the carbon atom of intermediate **c** to form intermediate **e**, which suffered intramolecular nucleophilic cyclization to produce **f**. Finally, the dehydration of **f** yielded substituted BT products **111**.

The same series of BTs **111a**–**j** were obtained in 42–99% yields by an effective cyclization of substituted ATPs with DMF in the presence of B(C_6_F_5_)_3_ combined with atmospheric CO_2_ (Figure 93). The outgoing intermediate dimethylamine in **b** further reacted with CO_2_ and silane catalyzed by B(C_6_F_5_)**_3_**, forming trimethylamine, which drove the cyclization reaction to the right, promoting the reaction [202].

A scalable CO_2_-mediated synthesis of BT **111a** in 86% yields was developed. ATP was cyclized with DMF as the carbon source in water as the solvent in the presence of CO_2_. The CO_2_ used could be recycled (Figure 94) [203].

Chun et al. reported the synthesis of an extended series of substituted BTs **112a**–**n** in 33–84% yields from substituted ATPs and CO_2_ using poly(3,4-dimethyl-5-vinylthiazolium) iodide as a precatalyst, 1,8-diazabicyclo[5.4.0]undec-7-ene (DBU) as a base, and phenylsilane as the reductant to in situ generate *N*-heterocyclic carbenes **a** (NHCs) by deprotonation (Figure 95), which capture CO_2_ to be transferred to phenylsilane as the formiate group in intermediate **c**. Then o-ATP forms the formyltioester **d**. The reaction was successfully carried out under mild conditions (1 atm of CO_2_ and 60–70 °C), with a broad substrate scope and functional group tolerance. The precatalyst salt was recovered and reused several times without any loss of activity [204].

## 3. Conclusions

Investigation into the synthesis of BT derivatives has increased in recent years. For example, in the past 5 years, we found 19 article reviews related to the strategies of synthesis and biological activities of BTs, of which 8 were reported in 2020. It shows the importance of the BT core in the pharmaceutical chemistry field. These works included only a few methods on the condensation of ATPs with carboxylic acids and their derivatives. Accordingly, in our article review, we have reported an investigation of about 20 years on the aspects of the research progress related to the condensation of ATPs with carboxylic acids and their derivatives, such as acid chlorides, amides, esters, orthoesters, nitriles, and thioesters, including carbon dioxide (CO_2_), as starting materials.

In this review, we found that the condensation of ATPs with carboxylic acids in the presence of PPA as the catalyst is still popular, although high temperatures, from 110 to 220 °C, and reaction times from 0.5 to 18 h are required. The best yields, of 80% to 90%, were obtained on heating at 140 °C for 24 h. These condensations have been carried out using other catalyst, such as P_2_O_5_/MeSO_3_H, PPA/H_3_PO_4_, trimethylsilylpolyphosphate ester (PPSE), MeSO_3_H/SiO_2_, triphenylphosphine (Ph_3_P), triphenyl phosphite (TPP), Samarium(III) triflate, trifluoroacetic acid (TFA), and iodo (I**_2_**). In these cases, the yields were from 60% to 99% but less reaction times were required. Few examples were found in which 80% to 90% yields resulted on heating in ionic liquids (10 min), on direct heating (16 h), as well as on refluxing in high-boiling-point solvents (8 h).

The condensation of ATPs with esters on heating under acid catalysts such as PPA, K_2_CO_3_/NMP, and AlMe_3_; under stirring; under direct heating; or in refluxing solvents depending on the ester has been carried out leading to low (14%) to high (92%) yields.

Condensation with orthoesters uses catalysts such as H_2_SO_4_, Bi(III) salts, supported fluoroboric acid HBF_4_-SiO_2_, BF_3_.OEt_2_, *o*-benzenedisulfonamide, and NH_4_Cl, with good yields, from 75% to 97%.

Acid chlorides and a nitrile-substituted ATP were condensed in refluxing chlorobenzene for 70 h under a stream of nitrogen to give the product in a 35% yield. The same condensation with unsubstituted ATP in refluxing chlorobenzene for 3 h gives the condensation products in 73–79% yields. On stirring in toluene for 15 min or 1 h at room temperature, the condensation yields were 50–98%. On refluxing in acetic acid, the yields increased to 70–88%. However, using *n*-methyl-pyrrolidone as a solvent on heating at 100 °C for 1 h gave the best results (82–95%). Ionic liquids such as 1-butylimidazolium tetra-fluoroborate ([Hbim]BF_4_) and 1,3-di-*n*-butylimidazolium tetra-fluoroborate ([bbim]BF_4_) as reaction media were used with excellent isolated yields. Pyridine as a base to trap the liberated HCl in the presence of *o*-benzene-di-sulfonamide as a catalyst has been used to produce BTs in 42–87% yields. On heating for 12 h at 100 °C with NaHSO_4_-SiO_2_ as a catalyst under solvent-free conditions, the condensation yields were 86–89%. The CuCl_2_/K_2_CO_3_ and KF/Al_2_O_3_ systems were excellent heterogeneous catalysts on refluxing in an adequate solvent to give 70% to 97% yields in this condensation. Acid fluorides attached to PEG resins were used, but poor isolated yields were obtained in the condensation (22–37%).

Condensation with nitriles produced moderate yields on direct heating with solvents or with the use of catalysts such as PPA. On using a heterogeneous catalyst such as Cu(OAc)_2_ on refluxing in ethanol or trifluoromethanesulfonic acid, BTs in best yields (50–97%) were produced.

Amides were condensed with ATPs on refluxing in solvents such as butanol or toluene by short reaction times (20 min–3 h) to obtain 50% to 94% yields. The use of imidazolium chloride or tetrabromo-methane as catalysts produced BTs in 60–90% yields, but 8–24 h were required. Condensation with substituted formamides in the presence of Zn(OAc)_2_**∙**H_2_O as a catalyst gives BTs in up to 95% yield.

Condensation reactions using microwave irradiation is the green method of choice because this method reduces the reaction time (less than 1 h), with or without the use of solvents, with good to excellent yields (50–97%).

Only one example on the use of ultrasound irradiation in the condensation of aromatic and aliphatic carboxylic acids was found with very good yields (82–92%).

## Data Availability

Not applicable.

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
