# Peer review of "Benzothiazoles from Condensation of o-Aminothiophenoles with Carboxylic Acids and Their Derivatives: A Review"

_molecules, 2021, doi:10.3390/molecules26216518_

Round 1
Reviewer 1 Report
The review deals mainly with the synthesis of functionalized 2-benzothiazoles via condensation of o-aminothiophenoles with carboxylic acids or their derivatives. Several synthesis are reported but it is not clear for a reader the take out message of the review; a long list of different synthesis is reported without a critical analysis on the pro and cons of the different methodologies (tolerance to functional groups, conditions, etc). Some of these informations are reported in the conclusion section, but should be expanded also in the main text. In my opinion, this is the main problem of this review.
The mechanism for the formation of the product should be reported in the beginning of each section, when changing the reagent (carboxylic acid, ester...); for example it is clear the machanism in Scheme 1, but not the one with other substrates (esters,...).
The numbering of the compounds should be completely revised, since it is misleading. Compound 2 in figure 2 is different from compound 2 in Figure 3, and also from cound 2 in Scheme 1. This problem must be solved throughout the manuscript. If a compond differs form another only for a substituent, the compound should be named Xa,Xb, Xc...
The use of colors in the text is not necessary and should be avoided. In the Figures its can help the reader, but not all the heteroatoms should be colored: I suggest to use color for the atoms involved in the specific reaction or to color the different functional groups in different colors (for example it could be more useful to have the whole BT in a different color).
Author Response
1.- The review deals mainly with the synthesis of functionalized 2-benzothiazoles via condensation of o-aminothiophenoles with carboxylic acids or their derivatives. Several synthesis are reported but it is not clear for a reader; a long list of different synthesis is reported without a critical analysis on the pro and cons of the different methodologies (tolerance to functional groups, conditions, etc). Some of these informations are reported in the conclusion section, but should be expanded also in the main text. In my opinion, this is the main problem of this review.
Response.
The take out message of the review is commented in the Abstract and Introduction sections as follows:
Some synthesis are included in several article reviews, however, any review about specific condensation reactions of o-aminothiophenols with carboxylic acids and their derivatives were found in the literature. On the other hand, several recent synthetic methods are included in this review that are not included in other reviews in the last ten years. For instance, conventional methods with the use of acid and metal catalysts as well as recent green techniques as microwave irradiation, the use of ionic liquids and ultrasound-chemistry (US), which have proven to have many advantages compared with reported methods.
With respect to the critical analysis of some synthesis, we think the review describe a summary of the synthetic methods, the critical analysis are found in the article. However, some commentaries about critical analysis were included in the text
The pro and the cons of the different methodologies are commented in the conclusions section and included in the text
The tolerance to functional groups is evident in each methodology.
The reaction conditions are commented in each synthetic method.
2.- The mechanism for the formation of the product should be reported in the beginning of each section, when changing the reagent (carboxylic acid, ester...); for example it is clear the machanism in Scheme 1, but not the one with other substrates (esters,...)
Response.
The mechanistic pathway for the condensation of o-aminothiphenoles with carboxylic acids is explained in scheme 1. The mechanism with the carboxylic acid derivatives is similar to that of carboxylic acids, however some mechanism were included and/or commentaries about the differences were included at the beginning of each section.
3.- The numbering of the compounds should be completely revised, since it is misleading. Compound 2 in figure 2 is different from compound 2 in Figure 3, and also from cound 2 in Scheme 1. This problem must be solved throughout the manuscript. If a compond differs form another only for a substituent, the compound should be named Xa,Xb, Xc..
Response
The problem was solved numbering with roman numbers in the introduction section. The Arabic numbering begins in the results and discussion section.
4.- The use of colors in the text is not necessary and should be avoided. In the Figures its can help the reader, but not all the heteroatoms should be colored: I suggest to use color for the atoms involved in the specific reaction or to color the different functional groups in different colors (for example it could be more useful to have the whole BT in a different color).
Response
The colors in the text were eliminated.
We used a color for the whole BT structure in the figures and schemes
Reviewer 2 Report
Cruz and co-workers report a good-quality review focused on the synthetic procedures to achieve benzothiazoles starting from o-aminothiophenols.
The topic relevance is demonstrated by the extensive literature reports covering this type of transformations and this review could be an efficient tools for chemists as an overview of the recent trends, novel reactions and mechanistic aspects. The length and number of reports cited is appropriate and well described, giving a good tool to understand this area of research for the scientific community. Suitable examples are selected in the logical order in respect of the nature of the reagent and catalysts used.
The review is suitable for the journal scope, and could be considered for publication after some minor revision.
- English level is good, however extensive and careful proofreading must be done, in order to correct misspelling and mistakes throughout the manuscript;
- The authors should add this reference: Curr. Organocatal., 2017, 164 which also covers the synthesis of benzothiazoles ;
- The subchapters numbering 2.2.1-2.2.3 should be corrected as 2.1.1-2.1.3 (pages 4-12);
- The bond angles of many structures should be adjusted and corrected in the entire manuscript (e.g. Schemes 19-20);
- Although the authors commented many of the methodologies, critically evaluating strenght and week points, some more comments on the characteristics of the synthetic procedures reported in chapter 2.2 should be added, for instance: the authors should comment on the advantages in the use of ionic liquids at pages 12-13 (description of scheme 25); or for examples why in the methodology at scheme 28 the obtained products did not have the iodine substitution anymore.
Author Response
- English level is good, however extensive and careful proofreading must be done, in order to correct misspelling and mistakes throughout the manuscript;
Response
We revised all the text and include some suggested comments.
- The authors should add this reference: Banerjee, S.; Payra, S.; Saha, A. “A Review on Synthesis of Benzothiazole Derivatives” Curr. Organocatal., 2017, 4, 164-181. which also covers the synthesis of benzothiazoles ;
Response
This reference has been included as suggested
- The subchapters numbering 2.2.1-2.2.3 should be corrected as 2.1.1-2.1.3 (pages 4-12);
Response
The subchapters numbering was corrected
- The bond angles of many structures should be adjusted and corrected in the entire manuscript (e.g. Schemes 19-20);
Response
The angles of the structures were adjusted
- Although the authors commented many of the methodologies, critically evaluating strenght and week points, some more comments on the characteristics of the synthetic procedures reported in chapter 2.2 should be added, for instance: the authors should comment on the advantages in the use of ionic liquids at pages 12-13 (description of scheme 25); or for examples why in the methodology at scheme 28 the obtained products did not have the iodine substitution anymore.
- With respect to the critical analysis of some synthesis, we think the review describe a summary of the synthetic methods, the critical analysis are found in the respective article. However, some commentaries about critical analysis were included in the text.
- A comparative analysis about the different methodologies are commented in the conclusions section, however some commentaries were included in the text
- The follow comment was considered in the text: In the acid conditions, the iodo group is electrophylically substituted
- The advantages in the use of ILs were commented in the text.
- The tolerance to functional groups is evident in each methodology.
- The reaction conditions are commented in each synthetic method.
